



# The limits to large scale supply augmentation: Exploring the crossroads of conflicting urban water system development pathways

Jonatan Godinez Madrigal[1,2], Nora Van Cauwenbergh[1], Jaime Hoogesteger[3], Pamela Claure Gutierrez[1] & Pieter van der Zaag[1,2]

[1]Department of Land and Water Management, IHE Delft, Delft, the Netherlands
[2] Water Management Department, TU Delft, Delft, the Netherlands
[3] Water Resources Management Group, Wageningen University, Wageningen, the Netherlands.

*Correspondence to*: Jonatan Godinez Madrigal (j.godinezmadrigal@gmail.com)

**Abstract.** Managers of urban water systems constantly make decisions to guarantee water services by overcoming problems
related to supply-demand imbalances. A preferred strategy has been supply augmentation through hydraulic infrastructure
development. However, despite considerable investments, many systems seem to be trapped in lackluster development
pathways making some problems seem like an enduring, almost stubborn, characteristic of the systems: over-exploitation
and pollution of water sources, distribution networks overwhelmed by leakages and non-revenue water, and unequal water
insecurity. Because of these strategies and persistent problems, water conflicts have emerged, whereby social actors oppose
these strategies and propose alternative technologies and strategies. This can create development pathways crossroads of the
urban water system. To study this development pathway crossroads, we selected the Zapotillo conflict in Mexico where a
large supply augmentation project for two cities experiencing water shortages is at stake. The paper concludes that urban
water systems that are engaged in a trajectory characterized by supply-side strategies may experience a temporal relief but
neglect equally pressing issues that stymie the human right to water in the medium and long run. However, there is not a
straightforward, self-evident development pathway to choose from, only a range of multiple alternatives with multiple trade-
offs that need to be thoroughly discussed and negotiated between the stakeholders. We argue that this development pathway
crossroads can cross-fertilize technical disciplines such as socio-hydrology, and social disciplines based on hydrosocial
studies, which both ambition to make their knowledge actionable and relevant.

## 1 Introduction

Urban water systems around the world are vulnerable in the face of climate change and uncontrolled urban growth (Flörke et
al., 2018). This alarming situation poses a risk that may sever water security for billions of people (WWAP, 2019; UNESCO,
UNWater, 2020). Historically, water managers have implemented large supply augmentation projects as their main strategy,
despite the piling evidence of its shortfalls (Gupta & Van der Zaag, 2008; Gohari et al., 2013; Rinaudo & Barraqué, 2015;
Purvis & Dinar, 2020). Some of these shortfalls arise because some problems consist of more than just issues of disparity
between water supply and water demand laid out on a spreadsheet. A more nuanced analysis of urban water systems shows
that there are problems of over-exploitation and pollution of water sources, low-investment capacity due to leakages, non-



revenue water and low water tariffs to improve the water utility´s performance, sustainability, and equitable service, expressed in persistent situations where poor users face constant disruptions of the water service and poor water quality (Biswas et al., 2018). Adding new sources of water to the system will likely not solve these issues (von Bertrab, 2003; von

Bertrab & Wester, 2005), nor make urban water systems more prepared for future stressors and shocks (Leach et al., 2010; Di Baldassarre et al., 2018).

The preference for large supply augmentation solutions is based on the sanctioned discourse and vested political and economic interests linked to the hydraulic mission that started at the beginning of the previous century and still continues as the main water paradigm in many countries (Allan, 2003; Boelens et al., 2019). This discourse has set an urban water system

trajectory whereby water managers underestimate the potential of alternatives and trivialize negative social and environmental effects of large infrastructure (GWP, 2012). At a local level, water utilities are often risk-averse and tend to opt for solutions contained within their own legal framework and engineering capacities confines (Lach et al., 2005).

Therefore, water authorities and managers seem unlikely to change the course of the strategies and policies they have implemented. Against the continuation of this trajectory based on large supply augmentation projects, social actors have long

criticized this sanctioned discourse that conceals and neglects more complex issues that are more difficult to tackle in urban water systems (COMDA et al., 2019). In addition, affected communities have constituted grassroots movements and a strong opposition against the implementation of these projects (Godinez Madrigal et al., 2020). In some cases, these movements have been effective in delaying or even cancelling these projects (Ahlers et al., 2017; Godinez-Madrigal et al., 2020), while demanding the need to search for alternatives (Ochoa-García, 2013; Ochoa-García et al., 2014; Ochoa-García et al., 2015;

Ochoa-García & Rist, 2018).

In fact, there is a wide range of available alternatives to large infrastructure for securing urban water security (Larsen et al., 2017). However, their potential in local contexts is disputed precisely by the water managers who aim to implement large infrastructure. This has generated an impasse, whereby large infrastructure is stalled, but alternatives remain untested. We call this situation characterized by conflict and indecisiveness in troubled urban water systems a development pathway

crossroads, in which the actors in conflict will either define a new pathway or reinforce the current one. Overcoming this crossroads is of extreme importance since it will imprint long-lasting consequences for the urban water system in question.

We chose the Zapotillo conflict in Mexico as an ideal case study of a development pathway crossroads. This case is a 15 years-old intractable water conflict between two urban regions seeking to increase their water supply through a large dam and intra-basin water transfer, while communities fight not to be relocated and farmers want to protect their water reserves in

the donor region. Due to the resistance of the opposing actors, the infrastructure project has been indefinitely stalled, and the challenging actors are lobbying for the implementation of alternatives to the Zapotillo project in the two recipient urban water systems suffering water shortages, overexploited water resources and an increasing water demand.

An effective analysis of this kind of development pathway crossroads would require a transdisciplinary approach, since there are constant feedbacks between the narratives, framings and social capitals that actors push forward to mobilize a particular

pathway over others, and the technical and biophysical reality that constrains the range of options available to actors.



Therefore, contrasting disciplines such as socio-hydrology (a technical discipline) and hydrosocial studies (a social discipline) may find fertile ground to make their knowledge actionable and relevant to social actors.

In this paper, we aim to analyse if and how the concept of development pathway crossroads can contribute to cross-fertilizing socio-hydrology and hydrosocial studies. Since the Zapotillo case is considered an emblematic and highly

intractable water conflict in Mexico (Godinez-Madrigal et al., 2020), its development pathway crossroads analysis should inform and help understand other water conflicts in similar situations but different contexts.

The paper is organized as follows. First, we discuss the relevant literature on socio-hydrology and hydrosocial studies that aim at improving our understanding of the coupled human-water systems, and the knowledge gap to improve the cross-fertilization between both disciplines to enhance their social impact in urban water systems. Second, we describe the

methodology, which involved ethnographic techniques and the development of a water resources model to elicit participatory processes and test scenarios. Then, we present the results, and finally discuss the relevance of the case to the understanding of development pathway crossroads.

## 2 Integrating disciplines to understand the development pathways of water systems

With habitual news headlines of cities reaching tipping points and 'day zero´s' in their water systems (Maxmen, 2018),

academic articles and reports calculating future billions of people without access to water (Vorosmarty et al., 2010; Schlosser et al., 2014; Mekonnen & Hoekstra, 2016; WWAP, 2019), and the incorporation of water in the investments of commodities of futures due to the growing fears for its scarcity (Bloomberg, 2020), water professionals look to implement tried-and-tested strategies based on their "technical expertise and the professional cultures that have developed over decades in line with the dominant urban water management system" (Larsen et al., 2016: 930). However, what if those tried-and-

tested strategies have inadvertently contributed to the present situation?

In the past years, the understanding of hydrological systems has incorporated society as a system that co-evolves with hydrological systems to the point of switching the analysis from the hydrological cycle to the hydrosocial cycle (Linton & Budds, 2014). This new understanding has shed light on feedback loops between society and water systems which constitute repetitive cycles causing unintended consequences. For example, one of these cycles, relevant to this paper, is that of the

'supply-demand cycle´ (Kallis, 2008).

Most human systems are dependent on water for their proper functioning, but since some systems are incentivized to keep growing, their water demand tends to overshoot water availability even when new water supplies are added (Gohari et al., 2013). Therefore, following this strategy continuously makes the system dependent on developing new sources of water, and then, more vulnerable when these eventually fail. This is known as the 'reservoir effect' (Di Baldasarre et al., 2018).

Although the 'reservoir effect' has been documented in many cities throughout the world, it is still unclear "how diverse combinations of hydrological, technical, and social factors play a role in accelerating or mitigating the underlying feedback mechanisms [of the reservoir effect in different contexts]" (Di Baldassarre et al., 2018:621). Answering this question is of





utmost importance if we want water systems across the world to avoid development pathways that set water systems into pernicious cycles. However, perhaps an even more important question to ask is how can water systems transit from a

pathway characterized by the supply-demand cycle to an alternative one with, hopefully, better sustainable, and social outcomes?

Many natural and social scientists around the world acknowledge a need for a larger scientific effort to actively get involved in solving intricate water problems (Castree et al., 2014). Exponents of the highly technical discipline of socio-hydrology have expressed their aim at contributing to science-policy processes to achieve the sustainable development goals (Di

Baldassarre et al., 2019); while social scientists have expressed their objective in not only studying water conflicts, but also contributing to transform them (Zeitoun et al., 2019). Moreover, Lave et al. (2014) considers imperative that more scientists "combine critical attention to relations of social power with deep knowledge of a particular field of biophysical science or technology in the service of social and environmental transformation".

However, this willingness may be futile if transdisciplinary research continues to be discouraged (Krueger et al., 2016).

Scientific endeavour may be handicapped to exert social impact, as explained by Lane (2017), since science is increasingly characterized by a frenzy of over-production of publications and of brutal competition that disincentivize scientists to be more reflective and formulate research questions "beyond the confines of the simple cause-effect models that it so often espouses" (Lane, 2017: 93). Moreover, it is still unclear how the social and natural sciences can collaborate in advancing both the understanding of complex phenomena while addressing societal challenges, since both differ in their ontologies -

how reality is perceived-, and epistemologies -how can reality be apprehended (Wesselink et al., 2017). This still constitutes a knowledge gap and a source of debate in academia (Di Baldassarre et al., 2019).

As a way out of this conundrum, Wesselink et al. (2017) suggested a modular approach, where both social and natural sciences can complement each other rather than compete, by means of developing social narratives from case studies that can benefit the exercise of socio-hydrological modelling and recognizing patterns in those case studies (e.g., Srinivasan et al.,

2012). In addition, Miller (2008) proposed an epistemological pluralism approach in which the methodology would be negotiated among the actors involved. Lane (2017) urged to conduct 'slow science', as a departure of a kind of science that privileges impact factors, and instead produce knowledge that is useful for the objects of its research when they are allowed to 'talk back' and influence research questions and how to answer them.

A promising framework to conduct transformative research is that of development pathways (Leach et al., 2010). This

approach engages with both the actions of actors and the narratives that frame how the coupled human-water system works, and collectively define what would be a sustainable pathway to undertake. Therefore, this approach brings about the politics between the actors and their strategies, which summon different social capitals to prioritize objectives akin to their interests and their vision for the future. This partly explains path-dependencies and lock-in systems, since actors in power keep making decisions and investing in the same development pathway (more than a pathway, it rather may be seen as a

"motorway" say Leach et al., 2010), until they are challenged by conflicts or crises (Godinez Madrigal et al., 2019). Thus, to



change course, scientists may need to conduct "empowering designs", to broaden out issues and problems, and opening up the decision space (Leach et al., 2010).

The development pathways approach further internalizes the "essential [task] to recognize the roles of public deliberation and negotiation – both of the definition of what is to be sustained [explicit qualities of human well-being, social equity and environmental integrity] and of how to get there – in what must be seen as a highly political (rather than technical) process." (Leach et al., 2010: 5). Water systems and their internal feedbacks and patterns cannot be fully understood without a thick comprehension of human agency and the multiple actor/power configurations that shape society (Rauschmayer et al., 2015). In this way, following the approach of development pathways, socio-hydrology and hydrosocial studies can increase the deliberation and negotiation, and thus, the potential of exerting much needed change in urban water systems.

## 3 Case study and methodology

### 3.1 Case study

To facilitate the reading of the paper and bring some context, in this section we briefly describe the Zapotillo project and introduce the nature of the conflict. The Zapotillo project is in México and centers around a proposed water transfer from the Verde River Basin to the cities of León and Guadalajara. Figure 1 illustrates the project and the three regions involved. First, the donor region of the Verde River basin has an area of more than 21,000 km2 and is primarily located in the State of Jalisco; the basin has a semi-arid climate in the northern half with an average precipitation of 330 mm/year and a sub-tropical climate in the southern half with an average precipitation of 900 mm/year. Then, the recipient regions are the cities of Guadalajara and León, with a population of more than 4.5 million and 1.5 million inhabitants, respectively.

The water transfer would start at the Zapotillo dam in the Verde River basin. The dam is currently built at 80 m height with a storage capacity of 411 million m3, but the states of Jalisco and Guanajuato are lobbying to increase its height to 105 m (with a storage capacity of 990 million m3) to secure water for both Guadalajara and León. However, the reservoir would need to relocate hundreds of people. Since the project announcement, the dam-affected communities, and many farmers from the region fearful of possible water scarcity derived from the project have resisted the implementation of the project. This has created a 15-year-old conflict that has polarized sectors of the society and added numerous actors intervening in the conflict (Godinez Madrigal et al., 2020). Since 2013, the reservoir construction has been interrupted at the original 80 m high and it is unclear if the project will ever be finalized and implemented.

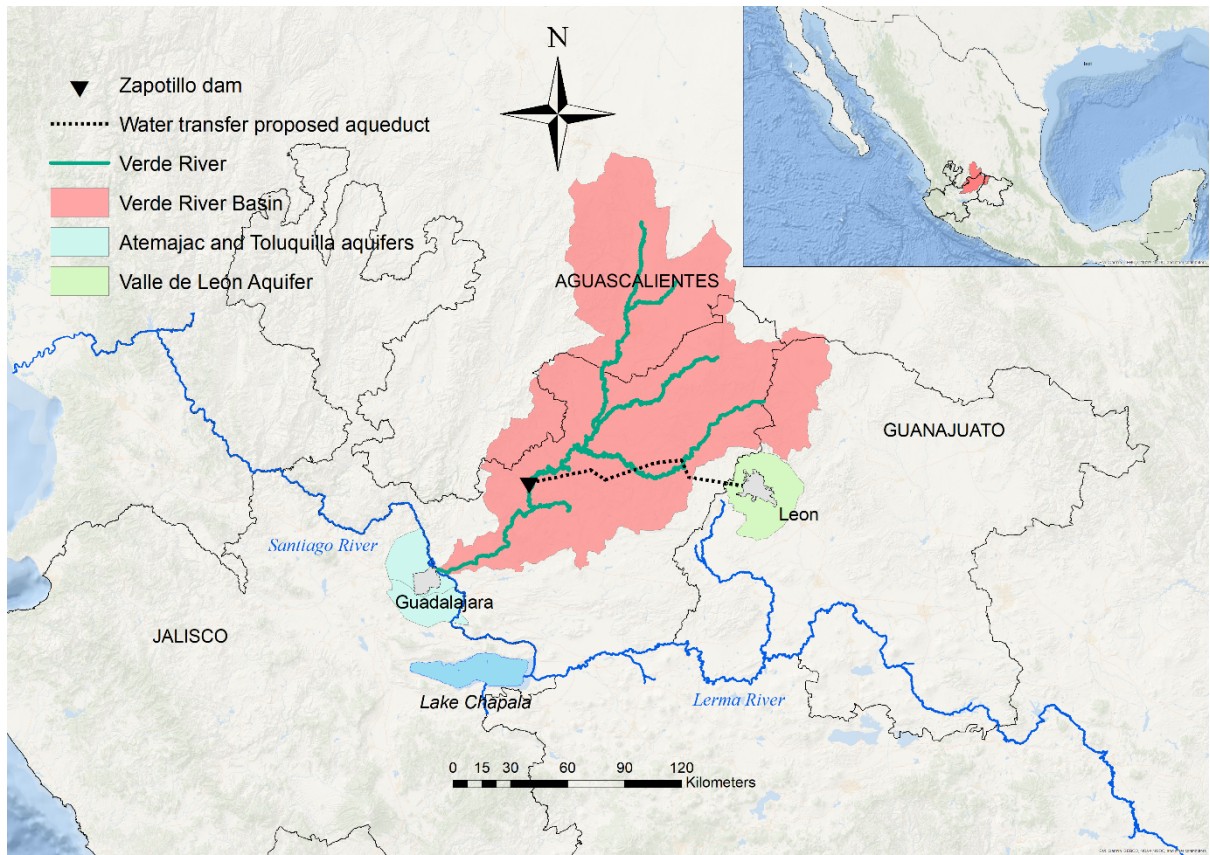

**Figure 1: Map of the Verde River Basin and main cities** (Source of GIS layers: © 2018 Conagua, and © 2019 Esri, Garmin, GEBCO, NOAA NGDC, and other contributors).

**3.2 Methodology**

Our research objective is to improve our understanding of what we have conceptualized as the development pathways crossroads of urban water systems. To achieve this, first, we ask what is the role played by hydrological, technical and social factors in a development pathway characterized by supply-side policies in Guadalajara and León, which would also contribute to the understanding of the reservoir effect (Di Baldassarre et al., 2018). Second, we ask how can water systems

transit from a pathway characterized by the supply-demand cycle to an alternative one with, hopefully, better sustainable, and social outcomes?

To answer the first question, we analysed the co-evolution of the water system and society in both Guadalajara and León to delineate what the current development pathways of these cities are. To answer the second question, we analysed how the decision space opened up to include alternatives to the current development pathway of Guadalajara and León; then, we

explored and analysed the new decision space, which considers the alternative development pathways proposed by some of the actors in the conflict. Our methodology is further explained in detail below.


### 3.2.1 Analysing the co-evolution of the human and water systems

First, we traced back the evolution of water use and population dynamics in both Guadalajara and León as far back in time as possible. For Guadalajara we found data since 1900, while for León data was available only since 1988. Then, we
normalized the data according to its initial value, following the same procedure as Di Baldassare et al. (2018).

To complement this analysis, we accounted for the political and socio-technical history behind these developments, a method inspired by the works of Kallis (2008) on the co-evolution in water resources development in Athens, and Molle & Wester (2009) on the longitudinal in-depth historic analysis of specific basins known as River Basin Trajectories.

### 3.2.2 Influence of conflicts in opening up the decision space

We conducted 22 in-depth semi-structured interviews with key actors in the conflict during the first half of 2017 and organized a stakeholder workshop with some inhabitants of Temacapulín in July 2017 to explore the root causes of the conflict. During that time, we also conducted participatory observation in meetings, forums, and other workshops to which the first author was invited until the end of 2019. We chronicled those meetings in fieldnotes which were commonly shared with the authors.

Then, to analyse the perspectives and perceptions of key actors to a larger decision space, by the end of 2018, the first three authors organized a participatory modelling workshop with the most important actors in the conflict in Jalisco. We invited representatives of Conagua (the national water authority), IMTA (the technical branch of Conagua), Jalisco´s government, Temacapulín and affected communities downstream, IMDEC (a prominent NGO working with the dam-affected communities of Temacapulín, Acasico and Palmarejo), Tómala (a civil society group involved in facilitating dialogue around
important societal challenges in Jalisco), college of civil engineers, and academics of local universities.

The workshop was structured around the interaction of the participants with a water resources model of the Verde River Basin (the donor basin of the Zapotillo project) and Guadalajara and León (the recipient regions) developed by the authors (Godinez Madrigal et al., 2018). The model, which was based on the model developed by UNOPS (Godinez-Madrigal et al., 2020) incorporated alternative urban water supply strategies to the Zapotillo dam, such as demand management, reallocation
of water rights and decentralized water supply augmentation (Supplementary material describes the model in detail). These strategies were previously proposed by the actors in conflict, but not yet formally developed in a water resources model. The model was controlled through a user-friendly interface developed in VBA, which we dubbed SimVerde (Craven, 2018; Godinez Madrigal et al., 2018). The actors randomly organized themselves in four groups to toy with the model, which allowed the generation of scenarios based on the discussion between the members of the group. To analyse the actors´
experience of a larger decision space through a boundary object such as the SimVerde, we debriefed the participants on their impressions of the workshop's experience on participatory modelling and how it changed their perspectives on the conflict.



Finally, in the last quarter of 2019 the first author conducted participatory observation in a series of workshops on alternative solutions to the Zapotillo project, one facilitated by the ministry of natural resources in Mexico City, and two others in Guadalajara organized by IMDEC.

### 3.2.3 Influence of conflicts in opening up the decision space

To analyse a larger decision space itself, we explored most of the combinations of the four main strategies mentioned before: (1) demand management, (2) reallocation of water rights (3) decentralized supply augmentation and (4) large centralized supply augmentation infrastructure. The first three strategies are composed of several measures, and the last one only of the Zapotillo project. The demand management strategy was composed of reclaimed wastewater for industrial water demand, implementation of water-saving devices, limiting urban growth, and reduction of physical losses in the distribution system. Reallocation of water rights was composed of water reallocation from agriculture to supply urban water demand. Decentralized supply augmentation was composed of implementation of rainfall harvesting systems and stormwater infiltration to different degrees. Finally, the Zapotillo project could be implemented by either of its four possible configurations (105 m dam, 80 m dam, 60 m dam, and decommissioning the dam; for details see supplementary material).

Our approach was informed by a 2019 stakeholders' workshop, where key actors explored a maximum of three alternatives. To make the exercise manageable, for each alternative we limited the number of possible combinations to one measure of each of the four groups (demand management, reallocation of water decentralized supply augmentation, and large supply augmentation infrastructure).

## 4 Results

### 4.1 The co-evolution of the urban water systems of León and Guadalajara

#### 4.1.1 León

Currently, León's water system appears to be in a dire situation. Local and national authorities recognize a severe over-exploitation of groundwater averaging a decline rate of 1.5 m/year (SAPAL, 2015). This level of overexploitation has had increasing negative consequences for the water quality of its aquifer (Villalobos-Aragón et al., 2012; Cortés et al., 2015). Despite this, León is the most economically vibrant city of Guanajuato producing 25 % of its GDP, partly due to its vibrant leather industry (Herrera, 2017: 86), and with the largest population, which grows at a rate of 2 % per year (Fig. 2). This constitutes a water challenge (or a dilemma), since this level of growth and groundwater over-exploitation seems untenable in the long term.



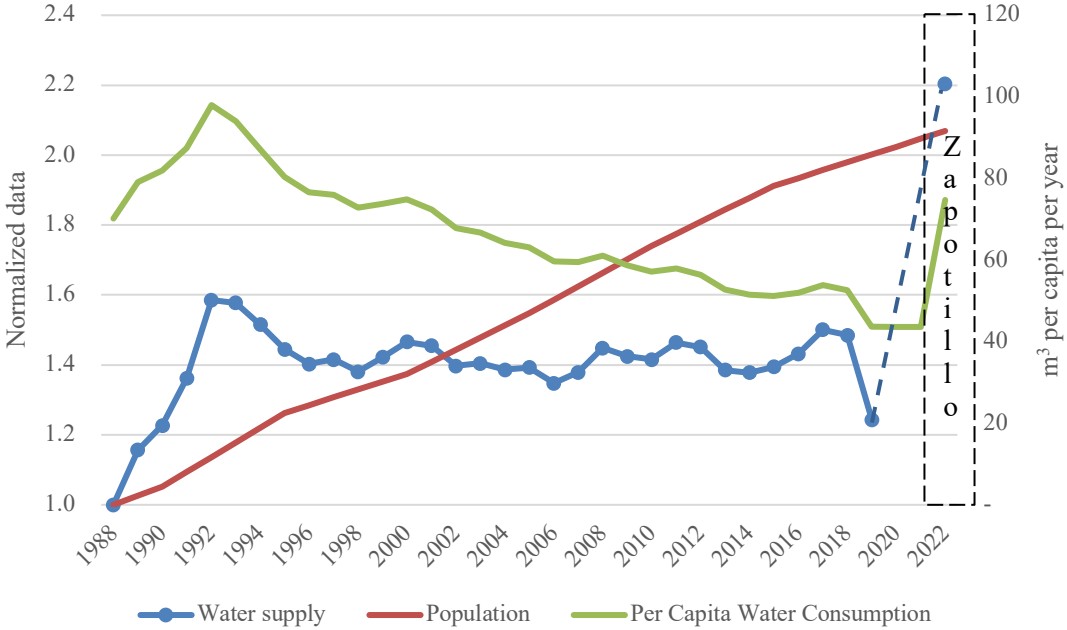

**Figure 2: The reservoir effect in León (the dashed line denotes the projected new water demand of a large new supply-augmentation scheme).** Sources of data: INEGI, 1990, 1995, 2000, 2005, 2010; 2015; CONAPO, 2015; CEA-Guanajuato & Conagua 2018; SAPAL, 2020.

Leon's water utility has focused its efforts on two strategies. One, depoliticizing water service to increase efficiency and achieve cost recovery since the early 1990s (Herrera, 2017). Two, since the 1990s, lobbying for a supply augmentation solution to Conagua, the national water authority (Godinez Madrigal et al., 2020). With these two strategies Leon's and Guanajuato's authorities have aimed at solving what they think is the limiting factor for their economic development: water scarcity (Rodriguez, 2008; Herrera, 2017; Pastrana et al., 2017).

The first strategy started after 1988, at a crucial point where the water utility (run by the municipality) had more than 60 % non-revenue water, and only 37% of its users enjoyed daily water service while poor neighborhoods would suffer no water service for days (Herrera, 2017). The system was in shambles since the water utility could not invest in infrastructure due to its below cost recovery water tariffs. These were kept low to avoid a politically costly decision of raising tariffs.

Since business organizations considered this situation a limit to their expansion and were the best organized civil society associations in León, they decided to organize a political opposition against PRI (the then ruling party). They found a powerful social cause and political flag in the improvement of León's water system. With this platform they gained powerful political traction to force the democratization of the city and force PRI out of the municipality and the water utility (Herrera, 2017). This would result in the decentralization of the urban water supply and the creation of SAPAL, a parastatal water utility.





The main objective of the newly elected water utility administration was to make the utility independent from politics by involving influential citizens as members of its administrative board and to consolidate the utility's efficiency. They aimed at reducing non-revenue water by means of renovating built infrastructure and achieving cost recovery by increasing water tariffs.

This strategy made León´s water tariffs the highest in the country and, consequently, per capita water use became one of the lowest in the country (Consejo Consultivo del Agua, 2011). This strategy was so effective in improving the utility's efficiency (reducing non-revenue water from more than 60% to less than 35% and water coverage improved to almost 95%), that Conagua awarded them with the prize of the best managed utility in the country in 2012. However, the social perception of the utility´s price hikes was that the utility was managed as a business when it should provide a public service (Caldera-Ortega, 2014; Lozano, 2014).

This strategy has also brought about unintentional consequences, since it led many industries, especially the small units of the leather industry, to resort to the black market, where they would buy water tankers from farmers engulfed by the city sprawl (Caldera Ortega & Tagle Zamora, 2017; Hernández-González, 2020). This also indicates that SAPAL´s (León's water utility) official water demand might be underestimated. Despite this, the policies of the city and the state have incentivized the formation of long-term large industrial clusters in the region by promising secured water supply (García-Garnica, 2017).

In early 2000s, this strategy alone proved insufficient for the objective of continued economic development. In 2004, SAPAL´s director mentioned that "[I]f we do not undertake a [supply augmentation] project in the coming five years […], we will not be able to have the same growth in León as we have today. We need to bring water, because we can still grow for five more years; afterwards, although we can sustain the supply to the city, we would need to halt its growth." (Rodríguez, 2004).

Therefore, in 2004, the national water authority announced the San Nicolas project in the Verde River basin, but a fierce and swift local opposition forced the authority to cancel it. In the same year, the water utility tried to acquire agricultural water rights from a neighbouring aquifer, but the local farmers resisted the project, ensuing a water conflict that ended with the cancellation of the project (Caldera Ortega, 2009). Finally, in the second half of 2005, the Zapotillo project was approved and publicly announced. However, since 2013 the project and dam construction have been stopped because of a social conflict of the dam-affected communities (Section 4.2 describes the conflict in detail). In response and to supply the growing water demand of the rapidly growing city, SAPAL has expanded its groundwater supply network to the aquifer of León as well as in the neighbouring aquifers of Silao-Romita, Turbio River and La Muralla. The number of deep tube wells of SAPAL grew from 124 in 2008 to 196 in 2019 and is pumping at ever increasing depths in all aquifers (Konijnenberg, 2019), in a context in which groundwater use for agriculture (accounting for up to 80% of all extracted groundwater) has gone by-and-large unchecked (Hoogesteger and Wester, 2017). In 2018, in view of the problems with the Zapotillo dam, CONAGUA declared the aquifer of Silao-Romita as having water availability (after years of deficit and declining water tables) to enable the institution to grant additional groundwater concessions to SAPAL (Hoogesteger, 2018). Additionally,





SAPAL has explored the possibility to use the small Palote Dam (now only used for flood control) for domestic water supply while also working on the further improvement of their distribution network, the re-use of treated wastewater for watering the green areas of the city, supporting the reduction of industrial water use especially the leather industry and stimulating a reduction in domestic water supply through public water awareness campaigns (Konijnenberg, 2019).

Although it is still unclear whether the project can be finalized, SAPAL and Guanajuato's government consider it the preferred solution to bring water security to León (SAPAL 2009, 2012, 2016, CEA-Guanajuato & Conagua 2018). However, it is still uncertain if the Zapotillo project will contribute to the sustainability of the system, since Guanajuato's water authority expects water demand to almost double when the Zapotillo project is implemented (see Fig. 2's dashed line) due to increased domestic and industrial water demand.

In conclusion, although León has implemented promising strategies regarding cost-recovery and reclaimed wastewater, they have been limited in scale, partly due to the expectancy of the Zapotillo project, which has disincentivized the implementation of upscaled demand management strategies (Caldera-Ortega et al., 2020). In the meantime, Leon's increased demand for water has been supplied through the expansion of the groundwater pumping network. Thus, even with the new water transfer the authorities do not expect a reduction of the groundwater overexploitation, since the 3.8 m3/s water transfer will not completely satisfy the new expected water demand of around 5 m3/s (CEA-Guanajuato & Conagua 2018).

### 4.1.1 Guadalajara

At the moment of writing this paper (2021), Guadalajara is suffering from a water shortage. The Calderón dam, a key water source contributing 14% of total water demand, is running dry and Jalisco´s politicians demand for the continuation of the Zapotillo project as the only solution (Del Castillo, 2021). This is the latest event of a controversial issue that has characterized Guadalajara for the past decades: how to secure the city with sufficient water supply for its increasing population of more than 4 million inhabitants (Godinez Madrigal et al., 2020). This issue is controversial because, although Guadalajara has benefited from three large supply augmentation projects in the past (Fig. 3), water managers have striven to continue adding even more water sources to the city (Flores-Berrones, 1987).




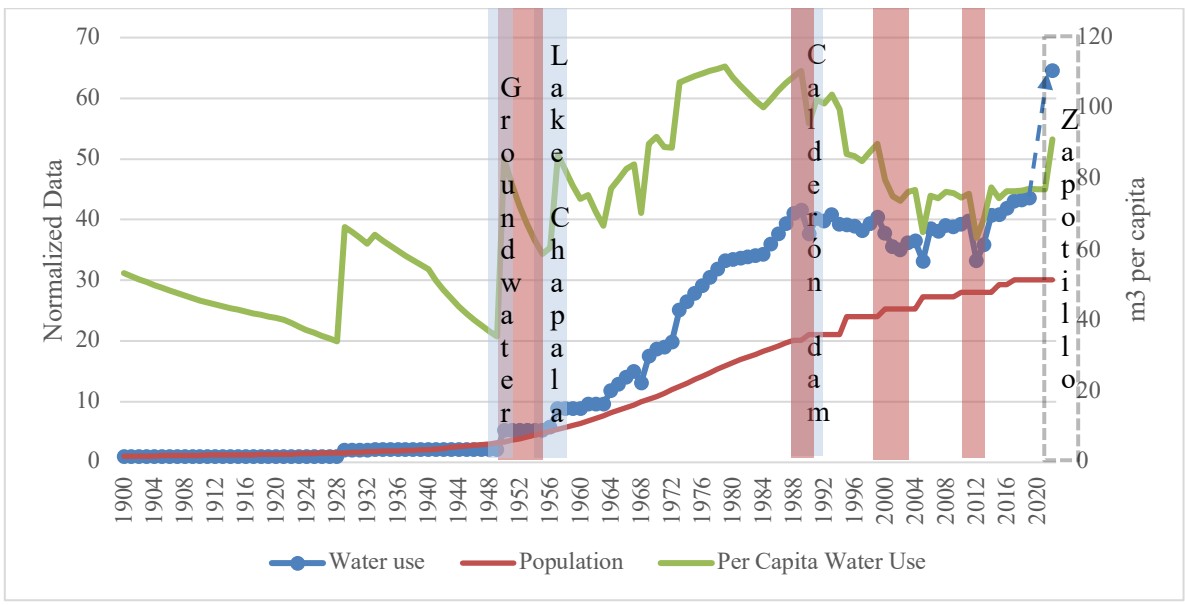


**Figure 3: The reservoir effect in Guadalajara (the blue bars denote the implementation of a large supply augmentation schemes in 1947, 1956, and 1991; red bars the presence of droughts; and the dashed lines the official projected new water demand for the proposed large supply-augmentation scheme).** Sources of data: INEGI 1900, 1910, 1921, 1930, 1940, 1950, 1960, 1970, 1980, 1990, 1995, 2000, 2005, 2010; 2015; Duran Juárez & Torres Rodríguez, 2001;
Jalomo Aguirre, 2011; Torres Rodríguez, 2013; Conagua 2015; SIAPA, 2014b; Gómez-Jauregui-Abdo 2015; CEA-Jalisco & Gobierno del Estado de Jalisco, 2018; SIAPA 2020.

The first supply augmentation project was implemented during the late 1940s and installed a network of wells to use groundwater that has continuously been expanded until today. That new water source increased water demand to around 280 l/d/cap (SIAPA, 2016). However, the accelerated population growth (higher than 6 %) typical of Latin-American cities of the
time (Camisa, 1972) and a severe drought that almost desiccated Lake Chapala (Godinez-Madrigal et al., 2019), generated a pressure to expand water supply. Therefore, in 1956 when the drought ended and the lake recovered, Guadalajara decided to build the Atequiza sluice in the Santiago River (which originates from Lake Chapala), that would become the main water source for the city. The city also built a large drinking-water plant with an installed capacity of 9 m3/s to increase water supply from the lake on demand. This infrastructure confirmed not only a dependency relation of Guadalajara to Lake
Chapala, but it also allowed urban growth and a higher water demand per capita (Fig. 3). Such growth was fostered by a combination of weak urban planning, and public policies to attract investment (Castillo-Girón et al., 1994).

During the 1980s, Guadalajara had already incorporated three adjacent municipalities as a unified, albeit chaotic, urban mass. Therefore, the authorities created SIAPA an intermunicipal water utility. SIAPA´s main mission was to expand water supply infrastructure to keep up with the urban expansion, rather than maintaining and improving the distribution network.





By the early 1980s the state and local governments increasingly developed groundwater, abstracting more than 2 m3/s of groundwater with 181 wells. Consequently, this level of abstraction started to present signs of over-exploitation in local aquifers. Therefore, in the late-1980s, an aqueduct directly from Lake Chapala replaced the Atequiza sluice, which was prone to surface contamination. And, projecting a continued water demand growth, water engineers of Jalisco and Conagua (the federal water authority) developed a basin development plan of the Verde River Basin, known as Zurda-Calderón, to

build more than 15 dams to expand Guadalajara´s water supply (Ochoa-García et al., 2014).

By early 2000s, Lake Chapala suffered again a water crisis that threatened 70% of Guadalajara's water supply that depended on Lake Chapala (6 out of 9 m3/s of Guadalajara´s water demand). SIAPA suspended water service in several parts of the city (Flores-Elizondo, 2016). To safeguard the lake, and more importantly, Guadalajara's main water source, Conagua mobilized resources and its authority to decrease the water use upstream in Guanajuato (for a more detailed description of

this event, see Godinez-Madrigal et al., 2019).

As consequence of the drought, in 2004, Jalisco spent millions of dollars in prospective studies for an intra-basin water transfer project from the Santiago River called Arcediano, which would supply as much as 10 m3/s to Guadalajara. The government had such high hopes that this project would solve the increasing water demand of Guadalajara in years to come that Jalisco´s government ordered SIAPA to grant any new housing water use request: "We can´t stop the city from

growing" mentioned a high-ranking civil servant (Del Castillo 2018).

The city´s dynamics (and many other cities in Mexico) has been characterized by a deregulated urban planning and a vigorous urban speculation (Pfannenstein et al., 2017, Reis 2017). This urban dynamic was further fuelled by cheap water tariffs set at lower than the cost of production. This situation undermined Guadalajara's urban water system, unable to invest in an aging and faulty distribution infrastructure (non-revenue water higher than 35% and most of its distribution system

reaching 80 years-old (Gómez-Jauregui-Abdo, 2015)). As a result, SIAPA was perceived as a water utility mainly managed to generate political gains rather than a good service based on technical and administrative sound decisions (Del Castillo, 2011).

Ultimately, the Arcediano project fell apart. Technically, the project was considered unfeasible due to geological complications (López-Ramírez, 2012). And, socially, civil organizations considered that the decision-making process

simulated public participation, lacked transparency, and broke political promises to constituents and stakeholders (Lopez Ramirez & Ochoa Garcia, 2012).

Jalisco´s government current position is that the water resources of the Verde River basin are the only solution to the water shortage problem and increasing water demand of Guadalajara. Should the Zapotillo project be implemented, the state water authority expects water demand to continue the growing trend from before the 1980s drought and reach almost 17 m3/s

(represented by the dashed bar in Fig. 3; CEA-Jalisco & Gobierno del Estado de Jalisco, 2018), and there will be no incentive to improve the current rate of non-revenue water caused by a lack of investment and low water tariffs.





## 4.2 Opening up the decision space

The past subsection shows that both cities´ urban growth has increased water demand beyond current water supply and for both cities the Zapotillo project is the only viable strategy to bridge that gap. Although León has focused on implementing some alternatives to supply augmentation, like reducing its non-revenue water to 32% and fostering reclaimed water for agricultural use, the continuing groundwater over-exploitation is their bottom line to lobby for the Zapotillo. Guadalajara has invested so many resources for the past three decades on developing the Verde River basin, that the Zapotillo project seems like the only option left.

However, a local network of social actors has lobbied to stop the Zapotillo project and proposed alternative strategies with fewer socio-environmental externalities (Godinez Madrigal et al., 2020). Although these actors have deployed a variety of legal, social, communicational, and political strategies to achieve these goals, they have only been successful in temporarily stopping the Zapotillo project by producing a cohesive narrative on how the Zapotillo will not fix the root causes of the problem. On the other hand, these actors have not been able to bring about viable and actionable alternatives.

Social actors developed a narrative consisting of how Jalisco's government has neglected the chaotic urban growth that has caused negative effects such as impermeable soils that obstruct groundwater infiltration and urbanized natural areas with high storm flow that are now causing recurring floods (CIESAS, 2018). In addition, they argue that SIAPA has not been able to further reduce its non-revenue water of 33% due to its obsolete leak detection methods, despite the local availability of advanced technology to swiftly localize them (Delgado-Aguiñaga et al., 2017). Related to water quality, water for domestic use does not meet drinking standards (McCulligh et al., 2020), forcing its population to depend on bottled water (Greene, 2018). Furthermore, most of its wastewater is untreated and currently affects the health of hundreds of thousands in the southern part of the city (McCulligh 2017), which in turn cannot make use of that surface water to alleviate water shortages due to its high concentration of industrial and domestic pollutants (UASLP & CEA Jalisco, 2010). Turning to León, experts and social actors have criticized that the utility has not been able to reduce its 32% of non-revenue water nor increase water reclamation for industries (Tagle-Zamora & Caldera-Ortega, 2021). Moreover, urban floods are common due to increasing surface impermeability, insufficient wastewater treatment that pollutes downstream ecosystems (Tagle Zamora et al., 2015), and a costly water service that privileges efficiency for the sake of economic growth over affordable water access to households (Tagle-Zamora & Caldera-Ortega, 2021). In conclusion, both actors from Guadalajara and León argue that the Zapotillo project does not consider nor aim to address the issues depicted above, because more water will not fix a broken system.

However, these criticisms do not directly translate into reliable alternatives, as the communities find it difficult to source sound expert opinions. The lawyer of the communities elaborated that: "A problem we often face is access to information; it has been opaque, not transparent and very technical […] another difficulty is to offer expert opinions; we have no one to do that. For example, a proof of environmental impact in anthropology, geology, engineering, etc. These are expensive [scientific] proofs. Sometimes we reach universities, colleagues, acquaintances. The communities cannot pay for those





expert opinions". Therefore, although alternatives like rainwater harvesting, limiting urban growth, reclaimed wastewater for industrial water demand, implementation of water-saving devices, and reduction of physical losses in the distribution system have been discussed on the media and the public agenda, they are yet to be developed into cohesive implementable plans due to lack of resources.

Water engineers further antagonized the actors against the Zapotillo project as only opposers without a constructive
criticism: "It´s so difficult to deal with these 'oposi-todos´ (anti-everything people)." (Anonymous interview with a retired water engineer of the state of Jalisco, 25 May 2018). This lack of development of alternatives has become a talking point for actors supporting the Zapotillo project: "Many organizations say they have been fighting for 10, 15 years for a position related to different alternative solutions for water [supply]. I think that if we continue like this for another 10, 15 years, then that method [sic] cannot deliver. It is just not possible to continue in the same situation for another 10 or 15 years."
(Transcript of a public talk of the Head of the civil engineer college of Jalisco, 22 Nov 2018). The head of the Water Council of Jalisco added: "If we would consider implementing these projects [i.e., reducing physical losses and rainwater harvesting], it would take years and be very costly" (pers. comm. 22 December 2020).

Considering these narratives, we found that a more quantitative approach was needed for a larger social impact, specifically in quantifying the potential of alternative solutions as viable pathways for the urban water systems of Guadalajara and León.
Therefore, the authors improved the UNOPS model of the Verde River basin by adding the other two concerned regions: Guadalajara and León. The model was designed to answer a critical question: how does the Zapotillo project compare to alternative solutions for creating a sustainable and socially just urban water system? In early December 2018, we organized a participatory modelling workshop in Guadalajara attended by representatives of seven organizations with opposite positions vis-à-vis the Zapotillo project.

Through a debriefing session, the outcome of the workshop can be summarized as follows: the participants chose to run scenarios based on their own interests. Social actors comprised of NGO and dam-affected communities sought for evidence that the Zapotillo project would not represent a solution; therefore, they tested scenarios with the Zapotillo infrastructure at 105 m without any alternative. Indeed, the model confirmed that scenario fared poorly in all its indicators (see Table 1). This group of actors sought to test the other 80 m, 60 m and decommission scenarios of the Zapotillo dam, but the model running
time (≈ 30 minutes) limited the number of scenarios to run within the workshop´s timeframe.

Participants from the national water authority sought to test the effect of alternatives to produce good results for urban water supply and showed scepticism of the model and the developed alternatives:

"You are trying to limit urban growth. I don´t think that is viable, I didn't understand that [measure] of limiting urban growth […] So, how are you going to make it happen [limit urban growth to 1 %/year?] It is unrealistic. If we are growing 2 %/year,
how am I going to decrease it to only 1 [%/year], sure not magically." (IMTA engineer, 6 Dec 2018).

Other participants acknowledged the uncertainties of the model as well and considered that citizens should also be part of choosing which data fed the model to increase trust. Moreover, the participant of Jalisco's government saw value in the model as a tool to aid in complex decision-making and celebrated an active role of scientists in controversial situations. And



participant engineers appreciated the capacity of the model to generate dialogue as a steppingstone for building concrete
alternatives:

"The importance of this methodology [participatory modelling] is that it has been able to generate dialogue without confrontations […] because [we] Mexicans do not know how to dialogue. We impose our vision, we want everybody to agree with us, and, this methodology, this process, generates its own structure to generate dialogue." (Former IMTA engineer, 6 Dec 2018).

Although all actors agreed that the model itself was incapable of finding an optimal solution to the conflict because of its inherent uncertainties and numerous configurations, most actors stated that participatory modelling could become a powerful process to engage actors to find negotiated solutions in the long run. Especially for communities affected by socio-environmental problems, the modelling tool seemed to have an additional potential:

I sincerely see this tool´s potential not so much for helping make decisions, but for understanding what the problem is. I was
envisioning... and felt emotional, that in my community we could have the chance to work the model with a lot of people. Because just imagine that the community could make a leap in understanding in a brief period of time a whole problem." (Member of an affected community, 6 Dec 2018).

This participatory modelling workshop led to follow-up activities. A year later some of the social actors who participated further explored alternative technical solutions by organizing a series of workshops in coordination with the newly elected
leftist federal government of Mexico, in which three alternatives were further explored with the assistance of a dozen of international and national experts in different fields: improving groundwater management, rainwater harvesting and reducing physical losses.

However, this series of workshops lacked the capacity to develop scenarios and compare them with the Zapotillo project. Therefore, to increase the social impact of scientific knowledge, we developed five sets of scenarios with different
combinations of infrastructure configurations. Table 1 presents the results and characteristics of the scenarios we explored. (See supplementary material for a more detailed description of these sets.)

**Table 1: Overview of the scenarios' performance.**

| Infrastructure configuration | | Dam | ⊗ | 60 m | 80 m | 105 m | 60, 80, 105 m |
|---|---|---|---|---|---|---|---|
| | | Alternatives | ✓ | ✓ | ✓ | ✓ | ⊗ |
| Number of scenarios | | | 15 | 30 | 30 | 30 | 3 |
| Indicators | > 95 % water supply coverage | | 0 | 8 | 8 | 24 | 0 |
| | Sustainable groundwater use | | 0 | 0 | 0 | 9 | 0 |
| | Meet Eflows requirements | | 15 | 30 | 30 | 15 | 0 |
| | Dam-affected communities are not flooded | | 15 | 30 | 0 | 0 | 1 |
| Number of acceptable indicators per group of scenarios | | | 2/4 | 3/4 | 2/4 | 3/4 | 1/4 |



The results of the fifth set shows that the Zapotillo project in its three configurations without the conjunctive use of
alternative strategies perform poorly, since they are not even able to produce a > 95 % coverage for Guadalajara and León,
which is the primary objective of the Zapotillo project. However, decommissioning the dam and implementing three
alternative solutions simultaneously cannot produce a > 95 % coverage for Guadalajara and León either.   Therefore,
decommissioning the dam might be counterproductive. On the other hand, the hybrid infrastructural configurations (using
the Zapotillo project in 60, 80 and 105 m in conjunction with the alternative solutions) present better outputs.

The different pathways represented by these infrastructural configurations show the multiple trade-offs involved with the
Zapotillo dam. On the one hand, with the group of infrastructural configurations with a 60 m dam, only 8 scenarios have a
positive > 95 % coverage for Guadalajara and León, but these scenarios cannot reverse the groundwater overexploitation
present in both regions. However, environmental flows meet the flows stipulated by the Mexican law (Norma Mexicana,
2012), and the dam-affected communities are spared.

The group of infrastructural configurations with an 80 m dam improves the groundwater indicator by helping reverse
groundwater overexploitation in León´s aquifer and curbs the overexploitation in Guadalajara´s aquifers. However, the
caveat is that, although they all meet the ecological flows in the Verde River, the communities will be flooded.

Finally, most of the scenarios of the 105 m dam group of infrastructural configurations have positive results in the > 95 %
coverage for Guadalajara and León indicator. Of these, a group of nine scenarios can reverse the groundwater over-
exploitation in both regions, and within that group, even three meet the ecological flows in the Verde River when the dam
management prioritizes ecological flows. However, all the dam-affected communities would be flooded. In conclusion, this
then is the crux of the crossroads: no alternatives exist that can satisfy all competing demands. Thus, trade-offs exist and,
inevitably, choices must be made.

## 4 Discussion

Socio-hydrology and hydrosocial studies are two (usually competing) disciplines that have focused on analysing the many
challenges facing urban water systems. There is a current heated scientific debate whether one discipline offers better
explanations or solutions for troubled urban water systems or if they can complement each other to make their knowledge
more actionable and relevant (Wisselink et al., 2017; Di Baldassarre et al., 2019). We argue that our concept of development
pathway crossroads offers a cross-fertilization opportunity for both disciplines.

Our retrospective analysis based on hydrosocial studies of Kallis (2008) and Molle & Wester (2009) of the co-evolution of
Guadalajara´s and León´s human-water systems shows that the trajectories of both cities have been defined by its continuous
and unrestrained socio-economic growth. As a result, these cities have sought additional water sources for at least the past
thirty years. Although water managers have warranted that quest for new water supply sources based on the "inevitability" of
socio-economic growth, we found that more than inevitable, socio-economic growth has been actively promoted as a
development pathway.





To fully understand the Zapotillo case, it needs to be framed under larger social, economic, and political dynamics. Therefore, although the project will certainly endow Guadalajara and León the necessary water resources to keep growing, it may not guarantee its long-term water security as explained below.

Our results show that the Zapotillo project is conceived by the authorities´ own accounts as a provisional strategy (CEA-
Guanajuato & Conagua, 2018; CEA-Jalisco & Gobierno del Estado de Jalisco, 2018), since they will require additional future large supply augmentation infrastructure once water demand outstrips water supply again in the coming decades. This is so, because there are socio-economic dynamics that are currently bounded by limited water availability, which would then be unleashed and be supported by an increased water supply. This behavioural pattern of cities and water managers is understood as the supply-demand cycle found in other cases around the world (Kallis 2008).

Furthermore, analysing the case of Guadalajara through the lens of the socio-hydrology work on the 'Reservoir effect' by Di Baldassarre et al. (2018), we found that the city´s current water shortage and its concomitant socio-economic damage (as of June 2021) is the result of the higher vulnerability of the city due its dependence to its current water transfers (Calderón dam and Lake Chapala). In turn, public pressure is being used by politicians to peddle the Zapotillo project (another water transfer) to shield Guadalajara from future water shortages. Thus, this decision continues the supply-demand cycle and an
increased dependency and vulnerability for Guadalajara in the future.

Figure 4 shows (in blue arrows) the composition of the feedback loops between the human and water systems that cause the "reservoir effect". However, since some of these feedback mechanisms are still not well understood (Di Baldassarre et al., 2018), we elaborated a more nuanced understanding of the variable 'public pressure' and introduce the concept of crossroads. We argue that once cities experience water shortages due to hydroclimatic trends and socio-economic growth,
politicians (and other economic interests) channel this public pressure to further and warrant large supply augmentation schemes, as in the present case of Guadalajara. However, grassroots movements, civil organizations, and scientists also device strategies to channel the public pressure to support alternative strategies and criticize the dominant large supply augmentation development pathway. This creates a rift in the decision space, which we have dubbed 'the development pathways crossroads' (shown in pink arrows in Fig. 4).



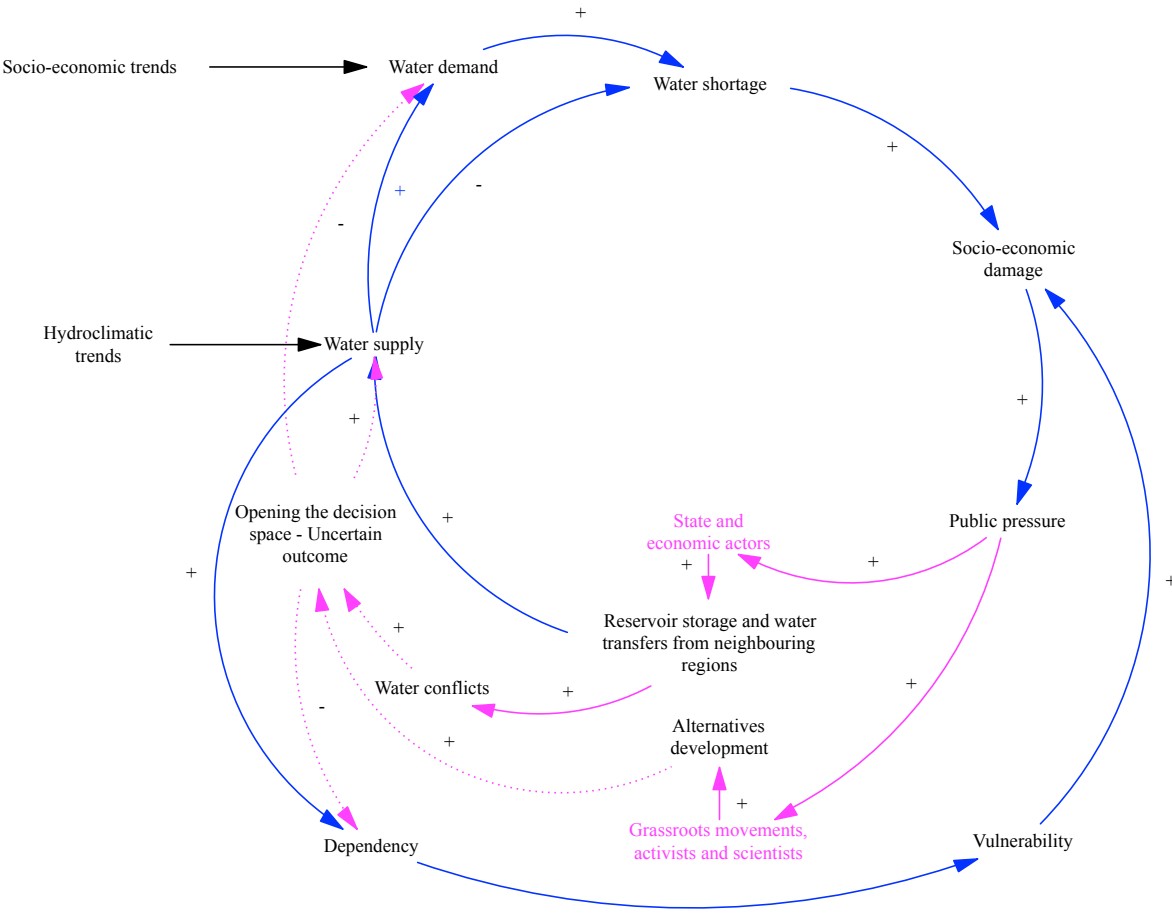

**Figure 4: The water conflict disruption in the reservoir effect (the dashed lines indicate their hypothetical status, while the pink lines indicate new variables not yet considered by the original conceptualization by Di Baldassarre et al. (2018).**

This kind of crossroads is characterized by conflict and a highly polarized public discussion packed with controversies of the subject, in this case, how to bring about sustainable water security to Guadalajara and León (Godinez Madrigal et al., 2020). Our results based on the hydrosocial approach show that the political economy of Guadalajara has subordinated urban water management as a function of urban expansion, while its water utility has not had leeway to set tariffs matching production costs to gather the necessary resources to maintain an aging and increasingly faulty infrastructure. The meagre resources have been directed to facilitate the addition of new customers, not increasing physical efficiency. Although León has been able to raise tariffs to maintain and improve its distribution network to a certain extent, its amassed financial resources have been saved to afford the Zapotillo´s water transfer as its only alternative to keep growing its economy.

Moreover, the development pathway characterized by continued population and economic growth has been naturalized as evidenced by the IMTA engineer ("So, how are you going to make it happen [limit urban growth to 1 %/year?] It is unrealistic.") The urban development of Guadalajara and León have privileged a chaotic urbanization that has disrupted





hydrological processes like stormwater infiltration, causing at the same time a lower rate of infiltration and recharge of over-exploited aquifers as well as costly urban flash floods. Therefore, analysing the Zapotillo case from the perspective of development pathways shows that events have a longer chain of causalities than is apparent at first sight. Not only is the conflict more than 15 years-old, but the social, political, and economic dynamics that have transformed the territory and made the Zapotillo project seem necessary dates decades back.

Likewise, the emergence of a competing alternative development pathway needs to be traced back since at least the inception of the conflict. However, transitioning to an alternative development pathway is usually faced with fierce opposition, since "[t]here is often assumed to be a singular path to progress, any questioning of which is taken to indicate an 'anti-innovation', 'anti-technology' or 'antidevelopment' stance" (Leach et al., 2010), and in the Zapotillo case also known as "oposi-todos". And critics often pitch a simplistic narrative of pitting the rights of the majority against the rights of the minority and ask the

latter to sacrifice for the "common good" (Roy, 1999, Leach et al., 2010). This narrative frames the minority as the culprit for not accepting the project, which is left unquestioned. In fact, Scott (1998) and Agrawal (2005) explain that 'high modernist' planning actively excludes political processes such as deliberation and negotiation precisely to avoid further questioning and the emergence of alternatives.

    Nevertheless, any kind of deliberation and negotiation around solutions for water problems in both cities has necessarily

been skewed, since the current decision space has been largely determined by the dominant large supply augmentation strategies. The technical actors promoting the Zapotillo project emphasized that there was not enough time and resources to develop any alternative solutions, since the water problems facing Guadalajara and León are so urgent that only the tried-and-tested, ready-made solution of the Zapotillo project is feasible. This professional culture of water engineers and managers acts as a powerful feedback mechanism that fosters the 'reservoir effect', since water managers consider that the

water problems facing Guadalajara and León are technical problems, and downplay the importance of deliberative and negotiation processes, which are political in nature.

    Our results of the scenario exploration analysis, a tool often used in socio-hydrology, show that even if an alternative development pathway is implemented, both urban water systems of Guadalajara and León will present difficulties in decreasing groundwater over-exploitation and achieving a higher than 95 % coverage. Therefore, if most optimal scenarios

require hybrid solutions (supply augmentation infrastructure plus alternatives), then both social actors and engineering and governmental actors need to acknowledge the necessity to open the decision space and include each other´s inputs, perspectives, and proposed solutions to build a sustainable and socially just urban water system.

    We argue that opening the decision space is an opportunity for socio-hydrology and hydrosocial scientists to contribute and complement each other's work to make it actionable and to increase its relevance. Some socio-hydrology research conclude

on the need to implement different strategies to increase sustainability and socially just outcomes in troubled water systems (i.e., Enteshari et al., 2020), but fail to acknowledge the socio-political complex dynamics of the water system, in which such strategies should be implemented. Such important contributions can fall on deaf ears and be ignored without the relationship with local actors.



Our experience from the participatory modelling process showed the importance of open science not only to replicate results
(Godinez Madrigal et al., 2020), but also to repurpose the design of a water resources model that was used to justify the
Zapotillo project by expanding its system boundaries. By adding Guadalajara and León water systems in the model to test
alternative water supply strategies (originally absent from the UNOPS model), the model could answer questions raised by
actors opposing the Zapotillo project. However, we were also limited by the many uncertainties inherited by the original
UNOPS model related to groundwater in the donor basin (see Godinez Madrigal et al., 2020), and the running time of the
model, which limited the number of scenarios ran by the stakeholders in the workshop.

The results of the participatory modelling workshop show that stakeholders are eager to learn and use new tools, but
especially to be considered in their design. Other actors even envisioned using the tool not for helping decision-making, but
for problem understanding within the affected community. Therefore, scientists can have an active role in promoting
deliberation and negotiation between stakeholders, even in contexts of conflict, polarization, and knowledge controversies.
Moreover, scientists may need to take on an additional role to repurpose their own tools to elicit the stakeholders' own
purposes and goals. As a result, a participatory modelling process can open the decision space by co-developing alternatives
proposed by actors representing a competing pathway to have a more balanced deliberation and negotiation processes even
in contexts of power asymmetry (i.e., Basco-Carrera et al., 2017; Van Cauwenbergh et al., 2018).

However, these roles may lie outside the current tasks that are expected from scientists (Lane, 2017), and may require an
active support from universities, since these social processes may not have immediate scientific relevance and require
investing time and resources (Lane, 2017). Furthermore, in the pursuit of more sustainable approaches, socio-hydrology
must not only aim at informing policymaking (Di Baldassarre et al., 2018), but also empowering actors that propose an
alternative development pathway through the co-production of knowledge and technical tools. Similarly, in the pursuit of
transforming conflicts, hydrosocial studies must not only challenge power asymmetries that seek to preserve the status quo
by means of identifying and describing power devices, but also identify and co-develop alternatives that can open the
decision space and boost the position and perception of social actors to articulate and enlarge a network of actors (Huitema
& Meijerink, 2010).

We are certainly not arguing that we have achieved a conflict transformation in the Zapotillo case nor found an optimal
strategy to bring about sustainability and socially just outcomes. Instead, we explored the complexity and difficulty of a
potential conflict transformation and the gamut of possible strategies through the lens of a development pathway crossroads,
which captures the tension between social and political actors with different visions analysed by hydrosocial studies, as well
as the potential future pathways studied by socio-hydrology.

**5 Conclusion**

This paper researched the current development pathway crossroads of the Zapotillo conflict in Mexico to understand the
multiple influences that determine dominant development pathways and the role of conflicts in opening the decision space to





embark a new alternative pathway for the urban water system. It did so by analyzing the urban water system trajectories that configured the present water scarcity and over-exploitation problems in Guadalajara and León and exploring the potential of alternative future pathways proposed by actors in conflict.

The dominant development pathway in Guadalajara and León has been characterized by a managerial, control-oriented
approach that went unchallenged for almost a century, what Leach et al. (2010) describe not as a pathway, but a 'motorway'. However, in the last three decades this pathway has been heavily scrutinized and thoroughly criticized by social actors opposing this kind of development pathway. This social opposition disrupted and caused two large infrastructural projects to fail and put the Zapotillo project in an indefinite hiatus. This hiatus has lasted 15 years, and to date it remains unclear which development pathway Guadalajara and León will undertake.

With a transformative spirit infused by the work of Di Baldassarre et al. (2019) and Zeitoun et al. (2019), we aimed at using socio-hydrology and hydrosocial studies to understand better the social, political, cultural, and economic factors and dynamics that have configured the development pathways at the crossroads and contribute to stimulate the necessary deliberation and negotiation of a larger decision space to explore a more sustainable development pathway.

Our research showed that the methodological framework of socio-hydrology related to the 'reservoir effect' (Di Baldassarre
et al., 2018), combined with the critical political ecology approach of hydrosocial studies (Kallis, 2008; Molle & Wester, 2009), can be used to problematize the still dominant sanctioned discourse of large supply augmentation infrastructure in other contexts. This exercise in conjunction with a participatory approach and an empowering design (Leach et al., 2010) can broaden what are the issues at stake in the urban water systems and open up the decision space beyond large supply augmentation infrastructure.

We broadened the issues by identifying that the main water problems are not only related to a discrepancy of water supply and water demand over time, but also to an unchecked and even sponsored economic and population growth, low water tariffs, aging distribution infrastructure and neglected intractable drivers like climate change. Our research design was also influenced by the actors who aim at finding alternatives to large infrastructure. With our water resources model and participatory modelling workshop, we opened up the decision space by modelling most of the alternative solutions brought
up by the actors opposing the conflict, the 'oposi-todos'.

We arrived at two main conclusions. One, if Guadalajara and León choose to follow the dominant development pathway, it is likely that they will trigger, and be trapped by, the 'reservoir effect' (Di Baldassarre et al., 2019) and make the urban water systems more vulnerable to intractable drivers of change in the future. And two, that although the dominant development pathway presented large drawbacks, we could not identify any alternative development pathways without trade-offs.
Therefore, if scientific disciplines like socio-hydrology and hydrosocial studies want to contribute to transform urban water systems to become more sustainable and just, they have to identify the hard choices that have to be made and to bridge the gap between technical and social disciplines to account for power relationships and the complex nature of water systems; as well as conducting 'slow science' (Lane, 2017) in close proximity with actors to contribute to more sustainable and just societies.



**Code Availability**

The reader can access the SimVerde software at: https://github.com/jongmadrigal/SimVerde.

**Data Availability**

The reader can access the data produced in the analysis of the different infrastructure configurations of the SimVerde at: https://github.com/jongmadrigal/SimVerde

**Author contribution:** Conceptualization, JGM, NVC and PvdZ; Data curation, JGM and PCG; Formal analysis, JGM; Investigation, JGM, JH, PCG and NVC; Methodology, JGM, NVC and PvdZ; Software, JGM and PCG; Supervision, NVC and PvdZ; Writing—original draft, JGM; Writing—review & editing, NVC, JH and PvdZ

**Competing interests**: The authors declare that they have no conflict of interest

**Financial support:**

This research has been supported by the Consejo Nacional de Ciencia y Tecnología (grant no. 409777).

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
