# Peer review of "The limits to large scale supply augmentation: Exploring the crossroads of conflicting urban water system development pathways"

_Hydrology and Earth System Sciences, 2021_

## Referee Comment (RC2)

**Revision Article:** *The limits to large scale supply augmentation: Exploring the crossroads of conflicting urban water system development pathways*

The article presents a very interesting case study that could provide a novel contribution to the discipline of sociohydrology. In particular, I think that there are sufficient qualitative and quantitative data to advance the concept of "supply-demand cycle" for instance by showing how this phenomenon is deeply intertwined with power dynamics produced and played out at various temporal and spatial scales. In addition, I think that there is space to show how these power dynamics can interfere with the cycle and in the long term, reshape the coevolution of water and society. I personally would like to see similar works published, therefore I encourage the authors to improve their analysis, in order to make their contribution clearer, more visible and therefore more significant. As of now, the paper is still a bit confusing. The authors use and merge together too many concepts, ideas and methods not always in a clear way. As a result, the paper reads more as a report of a professional consultant that touches upon different issues rather than a scientific publication that seeks to contribute to a specific (or more) body of knowledge. Ultimately, I suggest the authors to revise the manuscript so as to redefine what is their main message and their scientific contribution. I hope that my comments will be useful in this respect.

**INTRODUCTION**

1) **Confusing sociohydrological or socioenvironmental problem that this paper addresses**.
   At the moment I am still confused between the following issues:
   - The crossroads created within decision making processes that reshape urban water systems;
   - The preference for supply augmentation solutions relative to alternative pathways;
   - The supply-demand cycle and reservoir effects resulting from water supply augmentation projects.

   Even though you can mention all of them, I think that you have to make it clear what is the main issue that you are focusing on in this paper. Once you clarify this, I think that the research gap and the theoretical framework will become clearer to you and the reader.

2) **Unclear research gap.**
   You mention that you want to integrate hydrosocial studies and sociohydrology with development pathways concept but your paper does not show why this is needed. In other words, you do not explain what is the research gap that you are trying to address.
   I see a potential research gap in this paragraph: *"Although the reservoir effect has been documented in many cities….it is still unclear how diverse combination of hydrological, technological, and social factors play a role in accelerating or mitigating the underlying feedback mechanism."*.
   I think that your valid contribution to sociohydrology could start from here! But then you might want to choose which science provides you the best tools to contribute to sociohydrological literature (see following comments).

**THEORETICAL BACKGROUND**

3) **The literature section reads more like another introduction rather than a theoretical section.** I think that you cannot just briefly mention the theories you got inspired by and used. In my opinion you have to start engaging with them in a more nuanced way that is also more significant for your research (main issue: see comment 1 and 2).

4) **The paper is full of different concepts, ideas, and methodologies that belongs to different scientific scholarships,** i.e. development pathways, supply-demand cycle, reservoir effect, sociohydrology studies, hydrosocial theories, action research methodology. Even though I see the reason why the authors are referring to them, I fear that the way these concepts and theories are used and combined in the text is not being beneficial for the paper. It creates confusion and discrepancies in your arguments.

If the authors think that it is extremely important to use all these theories, concepts and methodologies together, they have to revise the article and find sufficient space in the text to motivate the use of these theories, concepts, and methodologies, to employ them in the case study, and lastly to show how the case study contribute to the body of knowledge selected. Personally, I would simplify, as the case study is already rich and interesting in itself.

5) **Sociohydrology and hydrosocial research are two different bodies of knowledge both focusing on human-water interactions.** However, they hinge on two different epistemologies, use different methods, provide different conceptualizations of society as well as of its interactions with water, and lastly, they sometimes reach different conclusions.
   In your paper this understanding is not clear. Also, at the moment it is still not clear why do you use them.
   I think that if you want to use these theories you might want to spend more time in explaining what they do and how one can enrich the other (even one paragraph would help).
   Probably it might be helpful to answer to the following questions: what is the main sociohydrological system that you are focusing on, what is the main sociohydrological process you want to study? How can sociohydrology help me to describe and study this system and its processes? How will hydrosocial studies enrich or change my understanding of this sociohydrological system and its processes? These questions will help you to define your scientific framework and contribution more clearly. And you might be better able to show (1) what do you use sociohydrology for (in your case you do that by explaining the supply-demand cycle and showing it with data) (2) what are the socio-political dynamics that you are exposing in your case, where do these dynamics originates (historically), and how they reshape human water interactions study at which scale.

6) **Limited literature review of main theories used.** Di Baldassare is indeed a prominent author and one of the first that contributed to the discipline yet beside the work of the reservoir effects (Di Baldassarre et al., 2018), there are also other works/authors that you might want to explore and eventually cite. (Di Bladassarre et al., 2017; Kuil et al., 2016; Garcia et al., 2016; 2020; Gohari et al., 2013; Madani and Shafiee-Jood, 2020; Sivapalan et al., 2012; 2014, etc.).
   I make the same comment for the hydrosocial researchers you cite. Besides Jamie Linton work (which indeed used the term hydrosocial for the first time). There might be other authors that discuss hydrosocial conflict, contestation, discursive construction of scarcity, etc. that would be a good fit to support your paper/case study. (Hommes, Boelens and Maat, 2016; Palomino-Schalscha et al., 2016; Boelens, Shah and Bruins, 2019; Budds, J., 2016; Lopez et al., 2019; Duarte-Abadía et al., 2015, Zwarteveen and Boelens, 2014; Zwarteveen et al., 2017; Swyngedouw, 2009, etc).

7) **You use the Leach et al. (2010) development pathways concept. But, you do not clearly explain why and what do you use it for.** (Is it for defining the main issue that you are describing in your case study? Is it to find a way to combine sociohydrology and hydrosocial studies? Is it to explain and justify your methodology?)
   I think that if you want to use it as lenses or main framework you have to further explain this concept and its theoretical background (especially for an audience that is not acquainted with the term) whilst at the same time also motivating the reason why you want to use it, the methodological choices it will imply and its limitations.

8) **Linked to the previous point: Melissa Leach is not a hydrosocial nor a sociohydrological scholar.** Her work fits better within the Socio Ecological Systems (SES) theory. In my view her ideas and theoretical framework reflect Elinor Ostrom political and institutional theories.
   How do you justify this choice? What is SES offering that hydrosocial theories or sociohydrology are not? I think that you have to choose which theory fits better (between sociohydrology, hydrosocial and socio ecological systems) to examine your case study. Or else be able to explain the added value of each theory on the examination of your case study. Again, the suggestion is to simplify and make it clear where do you stand.

9) **In your theoretical section you also mention transdisciplinary, yet this choice comes a bit as a surprise** especially considering that you do not define what do you mean with transdisciplinary, nor the reason why you need it. Maybe you should focus on that, instead of providing too much details that are not useful to your argumentation. Again, make a choice, prioritize and simplify.

**METHODOLOGY**

10) **Unclear case study:** In the case study you provide a very detailed description of the geographical characteristics of the area where the Zapotillo project will (or will not) take place. My first and minor problem with that is the inclusion of the detailed description of the case study before the methodology is been described. This is confusing and does not read well. In this section I will just mention that you use a case study, motivate the reason why you selected this case and detail the methods you are going to use to answer to address your research gap (qualitative, quantitative, action research etc.).
My second problem is that you seem to choose the Zapotillo project as a case study and then you perform a historical analysis of Leon and Guadalajara sociohydrological systems. I see where this come from (links are visible) but it creates confusion. I personally think that this confusion stems from the comment I wrote above in the Introduction section i.e. the fact that you do not define clearly what is the main focus of your work. I give you an example: If the focus was the supply-demand cycle, the case studies would have been Leon and Guadalajara. If the main issue was the (allow me the term) "Crossroads", then the case study would have been the Zapotillo project and the different conflicts generated by the project itself.
Try to clarify first the objective/focus and there redefine the case study.

11) **The methodology is described as a list of data used and activities performed**. I think that it would be much more useful to define the scientific method with something like: "In this paper we perform a inter or transdisciplinary exercise that merges qualitative and quantitative assessment of socioenvironmental processes in order to be able to understand and visualize the (long term) water-power dynamics etc…." Maybe here you could also explain the reason why you want to choose transdisciplinary rather than interdisciplinary or disciplinary methods.

12) **Is there a model/participatory modelling?** Even if you mention the use of a model, to me this model seems still a ghost, because it is there but that I cannot see it. I do not know what is the model about, is it about numerical modelling or scenarios development?
In my understanding, a (numerical) model is a numerical translation of a narrative or theory. Yet in your case I do not see any numerical model explained.
My suggestion is thus to simplify also here, using the right term (scenarios development?) and explain clearly in the methodology why you chose this method (participatory scenarios development?) and how you will perform and analyse the result.

**RESULTS**

13) **Is it a reservoir effect?** I think you have to be very careful with the term you use. If I am correct, reservoir effect occurs when reservoirs simultaneously secure water availability and increase the community's dependence on water infrastructure, resulting in higher vulnerability to, and impacts from, future droughts or water shortages. In your case more than a reservoir effect I see a supply-demand cycle. Both Leon and Guadalajara use different water sources for their supply: weirs, groundwater and dams, and not only reservoirs. I would encourage you to verify and be more precise in describing these phenomena.

14) **A bit more of political economy**? I personally liked the way you retraced the history of Leon and Guadalajara water supply and consumption alongside relating with the urban development of the two cities. I think that if this is going to be the main focus of your work you might want

to expand more on the political-economy of the two cities. Who benefited of these developments? Who lost? Since when these power dynamics are in place? Is there any legacy from the past? Is it worth mentioning this legacy?

15) **I think that a storyline (graph) would be nice**. Or you could even merge the water consumption and water supply graph with major historical decision, etc.

16) **I am not sure about the scenario development.** The way it is written is very chaotic, it includes too many details that are not very useful for the reader and it has no clear conclusions. My question for you is: are you really interested in the result or in the quotes that the participants said whilst you were doing the workshop? In my opinion these are the real results more that the workshop itself. Their quotes, ideas, tensions developed during the workshop might show you what are the power dynamics, what are the obstacles, who has more power, which alternative is possible or not and why. Personally, I am not interested in how you conducted the workshop (this could be placed in the supplementary material) but I am interested more in what the workshop was able to reveal about the local power dynamics! My suggestion is to rewrite it differently cutting and showing the qualitative data.

**DISCUSSION**

17) I liked line 477-480

18) **Issues with argument (1)**: I am not convinced about what you wrote on line 498-499. More than your argument this explain a power dynamic, otherwise known as discursive production of scarcity. Maybe you could use this to build up your arguments but not state that this is your argument.

19) **Issues with argument (2)**: I see and agree with the fact that there is a supply-demand cycle but I am yet unable to clearly see what are the additional power-dynamics that you have identified and the manner in which they modify the framework of Di Baldassarre et al., 2018.

20) **Issues with argument (3)**: Related to the above point I cannot understand figure 4 and I do not know if it is really useful as it is. Did you built this figure based on the workshop or based on your how qualitative and historical analysis? How do the power dynamics that you have identified change the water supply cycle, more or less water consumption, more or less water exploitation? Again, my suggestion is to simplify it and make sure that you show (more clearly) how certain power dynamics can modify the supply-demand cycle (show if they intensify them or not and explain how and why). At the moment I do not understand therefore I cannot agree with the updated causal loop.

21) **Issue with scientific contribution.** In my opinion it is still not clear what contribute to what. Is it hydrosocial research that helps unravelling power dynamics? Is it the Development Pathway concept that enrich sociohydrology? Or is it the use of action-research? I think that the authors might need to prioritise what contribute to what and how.

**MINOR COMMENTS**

- Line 27-28 In your statement it seems that all supply-augmentation projects are bad

- Line 128 Path dependencies and lock in: these two concepts are not introduced anywhere and it is unclear what you are referring to.

- Line 233: What do you mean with depoliticized strategy. It sound weird to me that they have used this word.

- Line 243 PRI acronym

- Line 265: Great quote.

- Line 294-296 The sentence/statement is unclear.

- Figure 3: Water use or water supply?

- Line 364 and 369: social actors is a bit vague

- Line 403: Here you start a completely different section for me. Suggestion is to either do another section or (what I would do) restructure this section by removing the detail of the workshop and including the quotes so as to show and highlight other power dynamics, actors, influence etc.

- Try to use simple words. I understand your willingness to be more sophisticated but it might be risky. Sometime I was not sure what you wanted to say, for instance line 11: lackluster, line 19: stymie, line 16: sever water insecurity; line 30: laid out on a spreadsheet; line 58: intractable water conflict; line 503 rift. My suggestion here is keep it simple.

- Figure 4: simplify revise and make cleared what are the effects of your new added line (see comment n. 20)

- The figures have errors and the line look sometimes confusing. Try to improve their quality, check the text.

- In general, English is ok but you can try to be even more straightforward.

Referee: savelli.elisa@geo.uu.se

---

## Author Response (AR1)

**Response to editor:**

We want to thank the editor for his constructive and swift review of our manuscript. We cannot thank enough the efforts of the reviewers and the editor to largely improve the quality of our manuscript.

**Editor comment:**

"I wish to thank the two referees who provided excellent, in-depth assessments of the manuscript. I also thank the authors for responding to the concerns and suggestions raised – a response that generally agrees to add additional material where suggested while largely rejecting the suggestions to remove material. With this approach the necessary focus (that both reviewers agree is needed) will not be achieved (given that the paper already contains much disparate material). Hence, I want to steer the authors towards greater focus which inevitably requires disconnecting with certain concepts that do not do much to advance the substance of the case study (another argument that comes out in the reviews)."

**Author response:**

The authors convened to accept the suggestions to remove material derived from the scenario exploration exercise, and focus instead on the decision space of the urban water systems elicited by the participatory modelling workshops. Also, we refocused our analysis towards the concept of crossroads, which led us to background some concepts we used from political ecology and socio-hydrology as described in detail in our second response.

**Editor comment:**

"I suggest you make the "crossroads" theme your central theme and bring in other concepts as auxiliary, but secondary concepts (I take it this is what you write in response to reviewer 2, point 4). If Leach et al. really is vital for your narrative then make it more prominent as a unifying framework, but at the expense of connecting with socio-hydrology for example. After all, you are analysing an interesting case from a critical water research perspective drawing on political economy (though this could be strengthened as per reviewer 2) and political ecology (which connects to the hydrosocial tradition but doesn't really require this as a concept). And you are connecting to phenomena discussed by socio-hydrology, but you can relate to these phenomena without connecting to socio-hydrology. Realising this will liberate you from some of the concepts and enable the necessary focus in research gap, contribution, theory engagement, case analysis etc. demanded especially by reviewer 2. I would also follow reviewer 2 in saying that the participatory modelling should not be a focus per se, but you should bring in this experience in describing the "crossroads" and controversy moments. Reviewer 1, too, suggests that the participatory modelling should perhaps be

another paper, so here you just need to draw on that experience for telling the story that has been emerging. In this light I would perhaps leave out Table 1. If Figure 4 is to stay, then the whole narrative needs to be developed towards this point."

**Response:**

The authors accept the suggestion of the editor of making the crossroads theme our central theme of the manuscript. We made other concepts such as the supply-demand cycle and the reservoir effect auxiliary to this end. In this way, we made more prominent the critical political economy perspective as suggested by both reviewers in section 4.1 of the manuscript. We also revised, improved and enhanced Section 4.2 on participatory modelling to better contribute to the story we are telling on the urban water system crossroads. To that end, we also removed Table 1 and our analysis of the scenario exploration exercise. However, we think Figure 4 can contribute to the crossroads story. Therefore, we decided to keep it, but, as suggested by the editor, redeveloped the text so that the narrative synchronises with it.

**Editor comment:**

"Instead of section 2, which opens connections to many concepts but doesn't do all of them justice (as the reviewers state), I would foreground the theory that is most vital for your narrative. In that respect, integrating disciplines perhaps isn't the right focus for the narrative."

**Response**:
The authors reconsidered the main focus of Section two. We acknowledged that due to the many concepts we were working on, we did not do them justice to foreground the research gap, contribution and theory engagement. Instead of integrating disciplines as the focus of Section 2, we realized that since our main theme was the crossroads, we needed to engage with literature that explains why systems remain without change and what could be the factors of change. Therefore, we foregrounded the role of water conflicts and grassroots movements as an underresearched theme that needs to be studied with a transdisciplinary approach.

---

## Author Response (AR2)

**Editor Comment:**

The authors have satisfactorily responded to the reviewer comments. There are just a few technical corrections left to be done:

Abstract:
Please explain notion of "development pathways crossroads".

**Authors' response:**

**We added a brief definition of development pathway crossroads in the abstract, line 16: "defined as a critical point whereby actors in conflict will either reinforce the current business-as-usual pathway based on large supply augmentation or implement alternative solutions for the urban water system.""**

**Editor Comment:**

Introduction:
Line 55, "to test this approach": Please define the approach you refer to here. But maybe testing an approach is not the right wording here.

**Authors' response:**

**We changed the wording as follows: "To test and analyse the concept of development pathway crossroads, we draw on empirical work…"**

**Editor Comment:**

Section 2:
Line 104: I don't think critical studies developed these concepts "to fill this gap in the natural sciences" – consider revising. You could use something like "meanwhile …" to contrast the previous paragraph.

**Authors' response:**

**We accepted the suggestion of the editor and replaced the sentence of line 106 with "Meanwhile, critical studies…".**

**Editor Comment:**

Methodology:
Line 131: Conducting an approach is probably not the right wording. Conducting a study? Taking an approach?

**Authors' response:**

**We changed the wording to "…we conducted an inter- and transdisciplinary study."**

L 139: I believe it is "participant observation".

**Authors' response:**

**Indeed, we corrected the term.**